# Relationship between the Accuracy of the Acetabular Cup Angle and BMI in Posterolateral Total Hip Arthroplasty with CT-Based Navigation

**DOI:** 10.3390/medicina58070856

**Published:** 2022-06-27

**Authors:** Hisatoshi Ishikura, Masaki Nakamura, Shigeru Nakamura, Takeyuki Tanaka, Hirotaka Kawano, Sakae Tanaka

**Affiliations:** 1Department of Orthopedic Surgery, University of Tokyo, 7-3-1 Hongo, Bunkyo-ku, Tokyo 113-8655, Japan; tanakata-ort@h.u-tokyo.ac.jp (T.T.); tanakas-ort@h.u-tokyo.ac.jp (S.T.); 2Department of Orthopedic Surgery, Toranomon Hospital, Tokyo 105-0001, Japan; mn0702@gmail.com; 3Department of Orthopedic Surgery, Nishitokyo Central General Hospital, Tokyo 188-0014, Japan; nakamura@med.teikyo-u.ac.jp; 4Department of Orthopedic Surgery, Teikyo University, Tokyo 173-8606, Japan; hkawano-tky@umin.net

**Keywords:** body mass index, computed tomography, navigation, obesity, posterolateral approach, total hip arthroplasty

## Abstract

*Background and Objectives*: Precise acetabular cup placement is essential for successful total hip arthroplasty (THA). In obese patients, its accuracy is often difficult to achieve because of the thickness of the soft tissues. This study aimed to determine the relationship between the accuracy of acetabular cup angle and body mass index (BMI) in posterolateral THA using the computed tomography-based navigation (CT-navi) system. *Materials and Methods*: We retrospectively reviewed 145 consecutive primary THAs using the CT-navi system between January 2015 and January 2018. All surgeries were performed using cementless cups employing the posterolateral approach with the patient in the decubitus position. We compared the radiographic inclination and anteversion obtained intraoperatively from the CT-navi with those measured by postoperative CT using three-dimensional templating software. We evaluated the relationship between the extent of errors and correlation with BMI. *Results*: In non-overweight patients (BMI < 25, 88 hips), the mean navigation errors for inclination were 2.8 ± 2.2° and for anteversion were 2.6 ± 2.3°. Meanwhile, in overweight patients (BMI ≥ 25, 57 hips), the mean navigation errors were 2.6 ± 2.4° for inclination and 2.4 ± 2.4° for anteversion. We found no significant difference between overweight and non-overweight patients in both inclination and anteversion. There was no correlation between the extent of errors and BMI. *Conclusions*: In posterolateral THA, CT-navi can aid the precise placement of the acetabular cup irrespective of a patient’s BMI.

## 1. Introduction

Total hip arthroplasty (THA) has been conducted for over 50 years; it is widely performed in patients with hip diseases. Because of the progression of implant design and the biomaterials used for bearing, THA has become one of the most common and successful surgeries [1,2] To guarantee a successful surgical outcome, the accurate positioning of the implant is vital. Acetabular component malposition is associated with instability, impingement, and accelerated wear, and may sometimes necessitate early revision [3,4]. Lewinnek suggested that the radiographic cup inclination should ideally be 40° ± 10° and anteversion should be 15° ± 10°, based on the rate of postoperative dislocation [5]. However, several reports have questioned the reliability of those results because dislocation occurs very often, even in the region of the Lewinnek’s safe zone [6].

Widmer et al. proposed combined anteversion (CA) to achieve a sufficient range of motion without impingement [7]. They proposed that the ideal CA, when the radiographic inclination (RI) of the cup lies between 40° and 45°, may be determined as follows: cup radiographic anteversion + 0.7 × stem antetorsion (SA) = 37.3°. Yoshimine et al. developed a model adding the cup inclination, which was given as the cup anatomic anteversion + cup RI + 0.8 × SA = 90.8° [8].

Past studies have demonstrated that the accuracy of the cup placement in THA without navigation is reduced in obese patients [9]. Moreover, the accuracy of the cup placement in obese patients also decreased in imageless navigation [10]. On the other hand, for the anterolateral THA in supine position, the accuracy of the acetabular cup angle using the computed tomography-based navigation system (CT-navi) was reported to be maintained irrespective of patients’ BMI [11]. However, to the best of our knowledge, there are no reports on the relationship between the accuracy of the cup angle and obesity in posterolateral THA using CT-navi. The accuracy of the cup placement decreases in the absence of CT-navi when the procedure is performed in the lateral position compared to the supine position in THA because it is more difficult to maintain its orientation in the lateral position [12]. In addition to posture, it is apparent that in posterolateral THA for an obese patient, the surgical view is deeper and narrower because of the thicker fat and gluteus maximus muscle.

Considering the scheme and the procedure of CT-navi, we hypothesized that we could minimize the effect of obesity during THA even with patients in the lateral decubitus position. Our study aimed to ascertain the relationship between the precision of the acetabular cup angle and the patients’ body mass index (BMI) in posterolateral THA with CT-navi.

## 2. Materials and Methods

### 2.1. Participants

We retrospectively reviewed 153 consecutive hips in which primary THA was performed using CT-navi from January 2015 to January 2018. Inclusion criteria were patients who underwent THA for osteoarthritis of the hip with Crowe type I [13], osteonecrosis, femoral neck fracture, and rheumatoid arthritis. In our institution, we used CT-navi for all patients who underwent THA. We excluded patients (8 hips) who experienced intraoperative issues such as apex pin loosening, mechanical seizure, and incongruity of verification. As a result, 145 hips were reviewed in this study. All the surgeries were performed by two orthopedic surgeons each with over 10 years of experience (M.N. and H.I.). Patients with a BMI of ≥25 were classified as overweight and those with a BMI < 25 were considered as non-overweight, according to the criteria of the World Health Organization (Figure 1). The retrieved patient data were anonymized and de-identified before the analyses. This study was approved by the institutional ethics committee of Teikyo University (No. 17-190: the approval date is 1 March 2018) and all the procedures were conducted in accordance with the Declaration of Helsinki and its later amendments. All participants enrolled in this study gave their written informed consent.

### 2.2. Preoperative Planning

The CT examinations were performed on three 64-detector row scanners (Light Speed VCT; GE Healthcare, Milwaukee, WI, USA/Aquilion; Toshiba Medical Systems, Tochigi, Japan/SOMATOM Definition Flash; Siemens AG, Forchheim, Germany) with a slice thickness of 0.5 mm. The CT images in DICOM format were transferred to both the three-dimensional planning software (Zed Hip, Lexi Co., Ltd., Tokyo, Japan) and the navigation planning station. Initially, using the three-dimensional planning software, the SA was determined to match the shape of the femoral medullary canal and to restore the hip center. If the native SA was too large (>40°) or too small (<10°), the modular stem was used to revise the SA. The cup RI was planned at 43° for all hips. The cup anteversion was obtained from those values and by calculation using the formula of Yoshimine [8]. Finally, we fine-adjusted the implant position, the length of the neck, and offset to strike a bilateral balance.

Furthermore, in the navigation planning station, we set several bony landmarks including the bilateral anterior superior iliac spine (ASIS), pubic tubercles and ischia, the pubic symphysis, and the sacral midplane to coordinate the pelvic position. The cup size and position were determined according to the plan using the three-dimensional planning software.

### 2.3. Surgical Procedure with CT-Navi

CT-navi (Stryker CT-based Hip Navigation System, Stryker-Leibinger, Freiberg, Germany) was used in all surgeries (Figure 2A). The posterolateral approach was used in all surgeries, with patients in the lateral decubitus position. In all the procedures, we used CT-navi without intraoperative fluoroscopy. Two apex pins were inserted in the ipsilateral ilium through small stab incisions. A pelvic tracker was affixed to the apex pins with an external fixation device (Hoffman II, Stryker-Leibinger, Freiberg, Germany) to enable detection by the infrared sensor camera. After exposing the acetabulum, initial paired-point matching registration was performed by digitizing the seven or eight landmarks including ASIS and the greater sciatic notch (Figure 2B). Surface matching of the pelvis was then performed by digitizing more than 30 points around the acetabulum and the ilium (Figure 2C). The reliability was verified by touching the bone surfaces, including both the inside and outside of the acetabulum, ASIS, and apex pin insertion site. During all the surgeries, we adhered to the preoperative plan as closely as possible. The cup angles on the display were recorded after achieving press-fit fixation of the cup (Figure 2D). The femoral stem was positioned without the navigation system.

### 2.4. Postoperative Evaluation

Postoperative CTs were obtained a week following the surgeries using a minimal radiation dose protocol [14]. The CT data were imported into the Zed Hip software for three-dimensional analyses. For postoperative measurement, the functional pelvic plane in the supine position was applied as the baseline for the pelvic plane. After matching the postoperative pelvis position with the preoperative one, the cup angles were measured. An experienced surgeon (H.I.) performed all the measurements, because the intraclass and interclass correlation coefficients for the reproducibility of measurements by the Zed Hip software were substantially high [15]. In this study, “navigation error” was defined as the difference between the calculated value, the value shown on the navigation display during the surgeries, and the one measured by the Zed Hip software.

### 2.5. Data Analysis

Statistical analysis was performed using IBM SPSS Statistics for Windows, version 24.0 (IBM Corp., Armonk, New York, NY, USA). The Student’s *t*-test was used to compare the mean navigation errors between obese and non-obese patients. The Spearman’s rank correlation coefficient was used to ascertain the correlation between the extent of navigation errors and BMI. The strength of the correlations, indicated by the Spearman’s rank correlation coefficient, were as follows: <0.3, negligible; 0.3–0.5, low positive; 0.5–0.7: moderate positive; 0.7–0.9: high positive; and 0.9–1.0: very high positive [16,17].

## 3. Results

Thirty-six patients were male, and 109 were female (145 hips). The mean age was 64.0 ± 10.1. The demographics of the two groups are shown in Table 1. The patients in the overweight group were significantly younger than those in the non-overweight group. As for the sex and diagnosis, there were no significant differences between two groups. Overall, the mean navigation errors in patients were 2.7 ± 2.3° for inclination and 2.5 ± 2.3° for anteversion. 

In non-overweight patients (BMI < 25, 88 hips), the mean navigation errors were 2.8 ± 2.2° for inclination and 2.6 ± 2.3° for anteversion. Meanwhile, in overweight patients (BMI ≥ 25, 57 hips), the mean navigation errors were 2.6 ± 2.4° for inclination and 2.4 ± 2.4° for anteversion (Table 2). There were no significant differences between overweight and non-overweight patients in both inclination and anteversion. The Spearman’s rank correlation coefficients were −0.04 (95% confidence interval, −0.2–0.13) for inclination and − 0.11 (95% confidence interval, −0.27–0.05) for anteversion, indicating that there was no correlation between the extent of the navigation errors and BMI (Figure 3 and Figure 4).

## 4. Discussion

This study indicates that a remarkable accuracy in cup placement can be achieved in THA using CT-navi (2.7° for inclination and 2.5° for anteversion). Moreover, CT-navi aided in the precise placement of the acetabular cup irrespective of a patient’s BMI.

Acetabular component malposition is associated with instability, impingement, and accelerated wear and may lead to early revision [5,18,19,20]. Many previous studies have described the effectiveness of CT-navi. A retrospective study demonstrated that fewer cups were placed outside of the safe zone and fewer postoperative dislocations occurred in the CT-navi group compared to the control group [21]. According to previous studies, the mean error of the cup angle with CT-navi was 1.5–3.0° for inclination and 1.2–3.3° for anteversion [22,23,24]. The result of our study is in agreement with those results.

Obesity makes the THA procedure more challenging. Coronal and sagittal pelvic tilt occurs frequently in the operating field and the surgical view is deeper and narrower due to heavy soft tissue. Previous studies revealed that obesity was associated with increased complication rates and poor outcomes following primary THA, including infection, dislocation, and early aseptic loosening [25,26]. Previous studies compared the incidence of these complications among several approaches. In THA in obese patients, there was no difference in the incidence of wound complications and deep infections when direct anterior and posterior approaches were compared [27]. Another study suggested that the increase in complication rates with THA for obese patients in comparison to non-obese patients was similar among several approaches [28]. Therefore, in THA for obese patients, the approach strategy is often based on the condition of each patient or the surgeon’s preference.

In THA without navigation, the accuracy of the acetabular cup angle can decrease in obese patients because of the depth of the surgical field and the narrow surgical view [9]. In THA with imageless navigation, the thick subcutaneous tissue causes the registration to be inaccurate, leading to the increased error of the cup angle [10]. On the contrary, in THA with CT-navi, the registration was performed by touching the bony surface directly. This procedure could help minimize the effect of being overweight in the posterior approach. The result of this study confirmed the usefulness of CT-navi, particularly in overweight or obese patients.

Several limitations must be considered when interpreting the findings of this study. First, the BMI of Japanese persons tends to be lower than those of persons from other countries [29]. Therefore, the number of obese patients (BMI ≥ 30) was relatively small (n = 14). Moreover, no morbidly obese patients (BMI ≥ 40) were included in this study. Further studies to investigate the relationship between morbid obesity and the accuracy of the implant positioning are required. Second, the pelvic coordinate system constructed on the navigation system used for surgery and the one constructed on the Zed Hip software used for postoperative measurement may not completely match. However, because both coordinate systems were constructed based on the pelvic bony surface and we uniformly used the radiographic angle for the measurements, the discrepancies should be minimized. With these limitations considered, the results of this study could be beneficial because the acetabular component accuracy in posterolateral THA with CT-navi was retained regardless of a patient’s BMI. In high-volume hospitals where CT-navi is used only for some patients, prioritizing obese patients would be one of the ways to use this resource efficiently.

## 5. Conclusions

In conclusion, posterolateral THA with CT-navi could offer precise placement of the acetabular cup irrespective of a patient’s BMI. However, further studies are needed to examine whether the accuracy obtained by CT-navi can reduce postoperative complications and improve the mid- and long-term survival of the implant in THA.

## Figures and Tables

**Figure 1 medicina-58-00856-f001:**
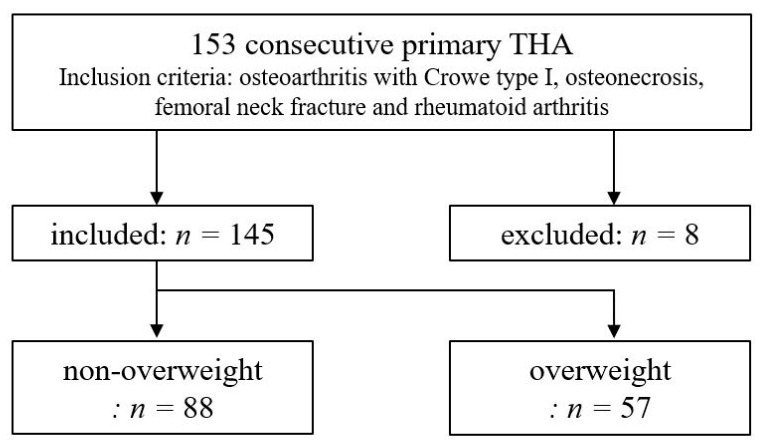
Study workflow.

**Figure 2 medicina-58-00856-f002:**
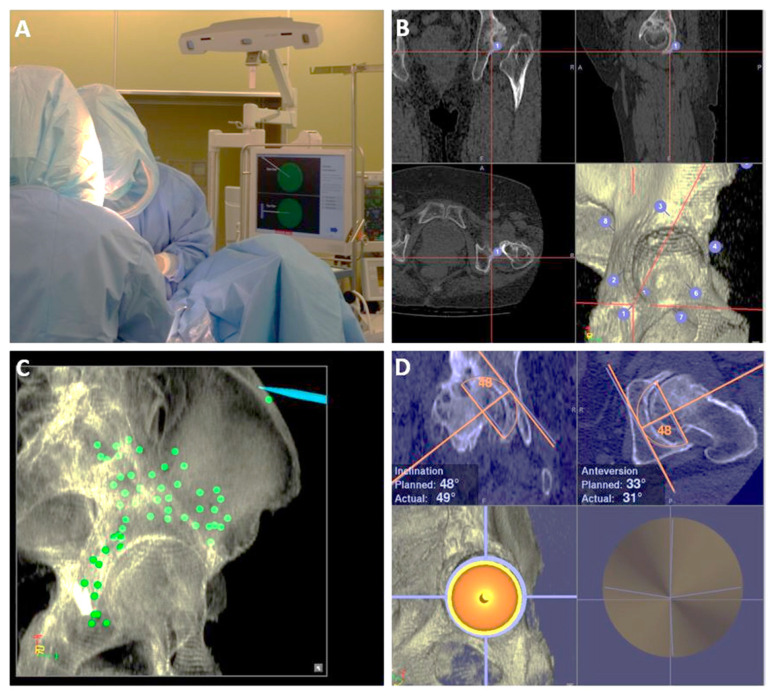
Intraoperative picture of CT-navi. (**A**) An overall picture of the CT-navigation system during the surgery. (**B**) Initial paired-point matching registration by touching several bony landmarks. (**C**) Surface matching of the pelvis by digitizing many points around the acetabulum and the ilium. (**D**) Cup placement during the surgery. Both the planned and actual angles are displayed.

**Figure 3 medicina-58-00856-f003:**
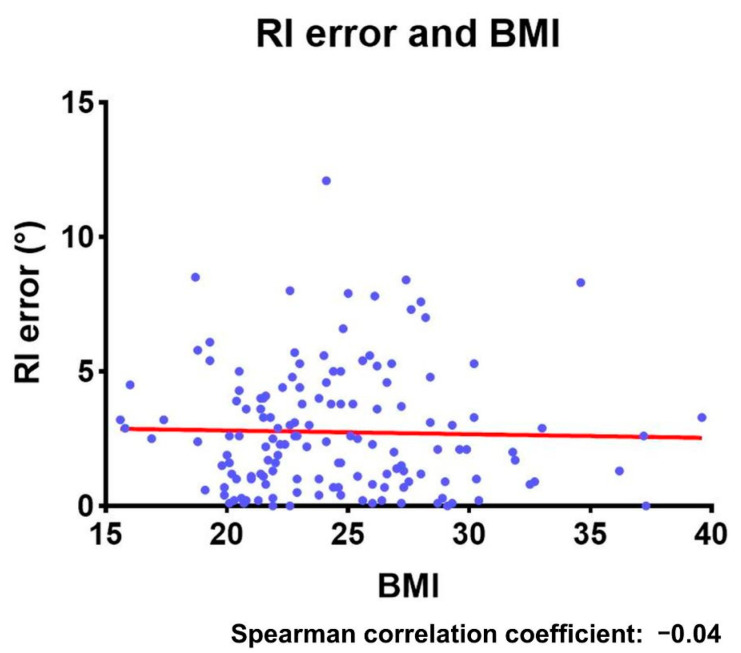
The relationship between the RI error and the patient’s BMI. RI: radiographic inclination; BMI: body mass index.

**Figure 4 medicina-58-00856-f004:**
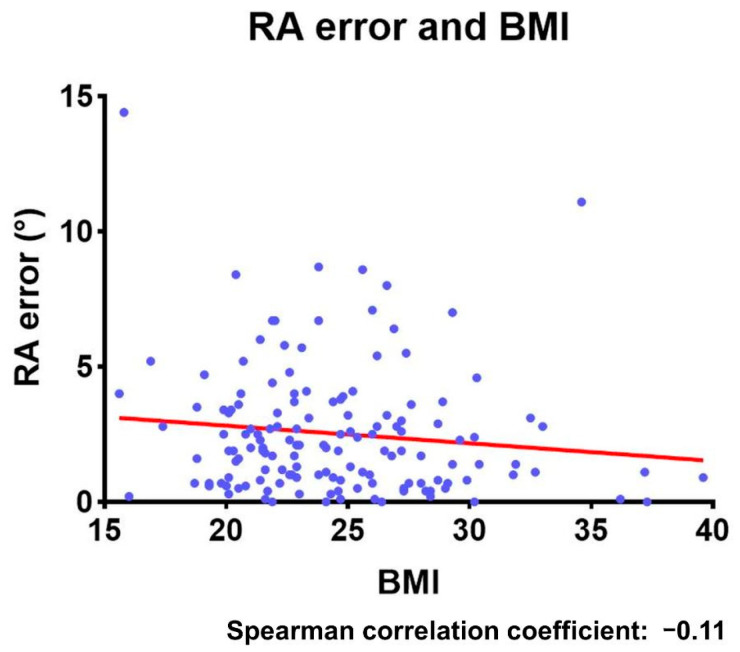
The relationship between the RA error and the patient’s BMI. RA: radiographic anteversion; BMI: body mass index.

**Table 1 medicina-58-00856-t001:** Patients’ demographic characteristics.

	Non-Overweight	Overweight	*p*-Value
Number of hips	88	57	
Mean BMI	21.7	28.7	<0.001 *
Age (years)	66.3 ± 11.3	60.5 ± 10.5	0.002 *
Female sex	70 (80%)	39 (68%)	0.13 **
Diagnosis			
OA	68 (77%)	45 (79%)	0.81 **
ON	17 (19%)	10 (18%)	0.91 **
FNF	2	1	
RA	1	1	

BMI: body mass index; OA: osteoarthritis; ON: osteonecrosis; FNF: femoral neck fracture; RA: rheumatoid arthritis * Student’s *t*-test ** Chi-square test.

**Table 2 medicina-58-00856-t002:** Navigation errors of the acetabular cup angle.

	Non-Overweight	Overweight	*p*-Value
RI (°)	2.8 ± 2.2	2.6 ± 2.4	0.67 *
RA (°)	2.6 ± 2.3	2.4 ± 2.4	0.58 *

RI: radiographic inclination; RA: radiographic anteversion * Student’s *t*-test.

## Data Availability

The datasets generated and analyzed during the current study are available from the corresponding author upon reasonable request.

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
