# Peer review of "Relationship between the Accuracy of the Acetabular Cup Angle and BMI in Posterolateral Total Hip Arthroplasty with CT-Based Navigation"

_medicina, 2022, doi:10.3390/medicina58070856_

Round 1

Reviewer 1 Report

We thank the authors of this article for the outcome comparison about the Relationship between the accuracy of acetabular cup angle and BMI.  The manuscript is generally well-written, however, there are notable deficiencies within the study, some of which were acknowledged by the authors.

          § Make your abstract more concise. The description of statistical methods can be excluded.

§ The hypothesis of this study is not detailed and unclear in the manuscript. It is the most important point to introduce this purpose to the authors. The conclusion is different from what we normally expect.  Previously published articles stating that cup position errors often occur during surgery on obese patients have not been introduced. Despite the lengthy explanation in the introduction, it is not clear what you want to describe within the purpose.

§ A description should be added for CT-navi. The overall picture, how to use it, whether there is any difficulty in applying it to obese patients, etc.  You need to explain the CT-navi which reference point was used for cup inclination and anteversion in the figures.

§ Please describe how you measured intraoperative stem anteversion (antetorsion) and how you calculated and determined the target anteversion of the acetabular cup.

§ Describe what reference point was used to measure anteversion and inclination on postoperative CT scan. For post-operative measurement,  add intraclass and interclass correlation as well as who measured it and how it was measured, it should be added.

§ You have to provide information about the surgeons who performed the operation. How many surgeons performed the operation, the surgeon's skill level, and the indication to THA using CT-navi should also be described.

§ In obese patients, coronal or sagittal pelvic tilt occurs frequently in the operating field due to heavy soft tissue. How CT-Navi can reduce the error should be described. Please add these issues to the discussion.

So please revise your manuscript to be convincing. In spite of these concerns, If the editor valued the subject in a good way, I agree with the editor's opinion. Thank you.

Reviewer 2 Report

1.      Line 2-3, the uppercase and lowercase for the title are not appropriate based on MDPI format, please revise it.

2.      Line 5-6, degree for the authors should not include, please delete it.

3.      The number of keywords is too much, please reduce it.

4.      Please reorder the keywords based on alphabetic order.

5.      What is the novelty of the present manuscript? It does not bring something really new since there are similar published literature has been published. The authors should highlight the novelty very seriously after revision. It is important for having novelty to give a worthy scientific contribution in this field, especially for total hip arthroplasty.

6.       In the introduction section, previous research with their findings and shortcomings has been explained by the authors but needs to extend for better explanation.

7.      Line 38-40 needs reference. Additional references published by MDPI should be adopted to support this explanation as follows: Computational Contact Pressure Prediction of CoCrMo, SS 316L and Ti6Al4V Femoral Head against UHMWPE Acetabular Cup under Gait Cycle. J. Funct. Biomater. 2022, 13, 64. https://doi.org/10.3390/jfb13020064

8.      Also, a sentence in line 40-42 and 42-44 have the same reference from number 1, the authors should change it to be different and prefer using from MDPI reference.

9.      Study workflow needs to present as an illustration to make the reader easier to understand rather than only dominant text.

10.   What is the bases/standard for selecting participants, it is not clearly explained?

11.   The selection criteria and inclusion procedure is not described properly. The patient involved is very important since impacting on the results presented.

12.   The number of involved patients is very small in number which makes the presented data not comprehensive. The authors need to care regarding this issue. Further explanation to support the use of a small number of patients is needed.

13.   Detail explanation of the patient involved based on group description should be provided.

14.   Line 186-199 is explained the study’s limitation, it should arrange in only one paragraph to make it more concise.

15.   The conclusion section has only one sentence, it is so wired. Please revise it, a more interesting conclusion with a solid explanation is needed.

16.   Further research needs to be stated in the conclusion section.

17.   The patent section should be deleted.

18.   English used still needs improvement, please proofread it, especially in English style.

19.   Please make sure the Medicine, MDPI template used was correct that used by the authors.

20.   Overall, the present manuscript is not giving serious scientific contributions a lack content. The substance is also very minimal that needs more additional data and explanation. Significance improvement is needed.

Round 2

Reviewer 2 Report

Well done to the authors for their work.